

# Optimizing alfalfa productivity and persistence versus greenhouse gases fluxes in a continental arid region

Jiao Ning[1], Xiong Z. He[2], Fujiang Hou[1], Shanning Lou[1], Xianjiang Chen[1], Shenghua Chang[1], Cheng Zhang[1] and Wanhe Zhu[1]

[1] State Key Laboratory of Grassland Agro-ecosystems, Key Laboratory of Grassland Livestock Industry Innovation Ministry of Agriculture, College of Pastoral Agriculture Science and Technology, Lanzhou University, Lanzhou, Gansu, China
[2] School of Agriculture and Environment, College of Science, Massey University, Palmerston North, New Zealand

## ABSTRACT

Alfalfa in China is mostly planted in the semi-arid or arid Northwest inland regions due to its ability to take up water from deep in the soil and to fix atmospheric N2 which reduces N fertilizer application. However, perennial alfalfa may deplete soil water due to uptake and thus aggravate soil desiccation. The objectives of this study were (1) to determine the alfalfa forage yield, soil property (soil temperature (ST), soil water content (SWC), soil organic carbon (SOC) and soil total nitrogen (STN)) and greenhouse gas (GHG: methane ($CH_4$), nitrous oxide ($N_2O$), and carbon dioxide ($CO_2$)) emissions affected by alfalfa stand age and growing season, (2) to investigate the effects of soil property on GHG emissions, and (3) to optimize the alfalfa stand age by integrating the two standard criteria, the forage yield and water use efficiency, and the total GHG efflux ($CO_2$-eq). This study was performed in alfalfa fields of different ages (2, 3, 5 and 7 year old) during the growing season (from April to October) in a typical salinized meadow with temperate continental arid climate in the Northwest inland regions, China. Despite its higher total GHG efflux ($CO_2$-eq), the greater forage yield and water use efficiency with lower GEIhay and high $CH_4$ uptake in the 5-year alfalfa stand suggested an optimal alfalfa stand age of 5 years. Results show that ST, SOC and RBM alone had positive effects (except RBM had no significant effect on $CH_4$ effluxes), but SWC and STN alone had negative effects on GHG fluxes. Furthermore, results demonstrate that in arid regions SWC superseded ST, SOC, STN and RBM as a key factor regulating GHG fluxes, and soil water stress may have led to a net uptake of $CH_4$ by soils and a reduction of $N_2O$ and $CO_2$ effluxes from alfalfa fields. Our study has provided insights into the determination of alfalfa stand age and the understanding of mechanisms regulating GHG fluxes in alfalfa fields in the continental arid regions. This knowledge is essential to decide the alfalfa retention time by considering the hay yield, water use efficiency as well as GHG emission.

Corresponding author
Fujiang Hou, cyhoufj@lzu.edu.cn

# INTRODUCTION

Alfalfa (*Medicago Sativa* L.) is the most widely grown perennial forage legume around the world (*Yang et al., 2008*). Since 2011, alfalfa planting has been gradually increased to meet the increasing demand for livestock production (*Wang, Hansen & Xu, 2016*). The increasing demand of alfalfa has great potential to change the structure and function of the farming systems but has positive and negative influences. First, alfalfa with rhizobia, the root-dwelling symbiotic bacteria, can fix atmospheric $N_2$ (*Peterson & Russelle, 1991*) and thus reduce N fertilizer applications. Secondly, due to its strong ability to take up water (*Wan et al., 2008*), the alfalfa is mostly (>75%) planted in the Northwest inland regions with an arid or semi-arid climate in or around the Loess Plateau of China (*Hu & Cash, 2009*; *Wang, Hansen & Xu, 2016*); however, alfalfa has a much higher water requirement than other crops which may deplete soil water and aggravate soil desiccation in long-term stands (*Guan et al., 2013*; *Zhu et al., 2016*). For example, *McCallum, Peoples & Connor (2000)* reported that in Australia soil profiles under alfalfa-based perennial fields remain consistently drier throughout the year compared with continuous annual cropping. After eight years of alfalfa cultivation on the north edge of Loess Plateau, China, four to five years are required to restore soil water condition to the initial level (*Du, Wang & Long, 1999a*), imposing a negative effect on the growth of subsequent crops. Moreover, the forage yield as well as the water use efficiency usually decrease after four to seven successive growing years (*Du, Long & Wang, 1999b*; *Zhang et al., 2004*; *Cheng, Wan & Wang, 2005*; *Jia et al., 2009*). Therefore, determination of the optimal cultivation ages of alfalfa is critical to avoid over-consumption of soil water by balancing the forage yield and water use efficiency.

Another concern with alfalfa is that the possible rhizobial denitrification may result in an increase of greenhouse gas (GHG, mainly the nitrous oxide $N_2O$, methane $CH_4$, and carbon dioxide $CO_2$) emissions (*O'Hara & Daniel, 1985*). It is well known that $N_2O$ effluxes are driven by nitrification (oxidation of $NH_4^+$ to $NO_3^-$ via $NO_2^-$) under aerobic conditions and denitrification (reduction of $NO_3^-$ to $N_2O$ and $N_2$) under anaerobic conditions (*Ussiri & Lal, 2013*; *Oertel et al., 2016*). $CO_2$ release from soils is the subsequent results of soil respiration of both root and anaerobic and aerobic microbes (*Oertel et al., 2016*), where root respiration may contribute average up to about 50% of the total soil respiration depending on the season and vegetation type (*Hanson et al., 2000*). $CH_4$ in soils is produced by methanogenesis under anaerobic conditions and is consumed by methanotrophic microorganisms that use $O_2$ and $CH_4$ for their metabolism under aerobic conditions (*Smith et al., 2003*; *Dutaur & Verchot, 2007*; *Gao et al., 2014*). Thus, whether soil is a net source or sink for $CH_4$ depends on the relative rates of methanogenic and methanotrophic activity (*Tate, 2015*; *Tian et al., 2016*). GHG fluxes are mediated by both biotic (e.g., microbial activity and root respiration) (*Kitzler et al., 2006*; *Singh et al., 2010*; *Butterbach-Bahl et al., 2013*) and abiotic (e.g., soil temperature, moisture, and soil carbon and nitrogen) factors (*Kitzler et al., 2006*; *Singh et al., 2010*; *Butterbach-Bahl et al., 2013*). Heretofore, how these factors influence GHG fluxes or which one is the most important factor influencing GHG fluxes in the arid continental regions is still not clear.

Previous studies on optimizing alfalfa stand age usually consider only two factors, the yield of alfalfa hay and water use efficiency, especially in the arid or semi-arid regions (e.g., *Zhang et al., 2004*; *Cheng, Wan & Wang, 2005*; *Fan et al., 2016*) but ignores greenhouse gas (GHG) effluxes. In contrast, other studies investigate GHG effluxes from alfalfa fields of different stand ages (e.g., *Zhong, Nelson & Lemke, 2011*; *Uzoma et al., 2015*; *Burger et al., 2016*) but do not measure alfalfa productivity; furthermore, those studies usually only consider $N_2O$ effluxes due to the $N_2$ fixation of alfalfa. So far only a few studies on GHG effluxes from alfalfa fields have included $CH_4$, $CO_2$ and $N_2O$ in analyses (e.g., *Chaves et al., 2006*; *Ellert & Janzen, 2008*).

In this study, we estimated the persistence of alfalfa in an arid continental region with respect to the tradeoffs between hay yield, water use efficiency, and GHG effluxes as affected by soil properties. The approach was to integrate the forage yield and water use efficiency with total GHG efflux ($CO_2$-eq) and GHG efflux ($CO_2$-eq) per unit hay yield. We then investigated the dynamics of soil properties (i.e., soil temperature, water content, organic carbon and total nitrogen), root biomass and $CH_4$, $CO_2$ and $N_2O$ fluxes during the growing seasons, and finally analyzed the influence of soil properties and root biomass on GHG fluxes. Results from this study will improve our understanding in GHG effluxes in the arid areas and provide essential information to develop strategies for alfalfa field management.

## METHODS

### Study site and alfalfa field

The study was carried out in Grassland Agricultural Trial Station of Lanzhou University (latitude 39°15′N, longitude 100°02′E), Gansu Province, China. The field ($\approx$ 280 ha) used in this study is 1,390 m above the sea level and classified as a typical salinized meadow with temperate continental arid climate in the Northwest inland regions (*Zhu et al., 1997*). The mean annual precipitation is about 123 mm with $\geq$ 65% occurring during the growing seasons from April to October (*Kobayashi et al., 2018*). Irrigation is necessary and usually applied bimonthly during the growing seasons with a rate of 120 mm respectively in April, June or August in the study site. The annual mean air temperature is 7.6 °C (from −28 °C between December and February to 38° C between June and Mid-August). The soil pH value is about 8.0, and the soil at the study site is classified as Aquisalids according to USDA soil taxonomy (*Zhu et al., 1997*).

To optimize the alfalfa stand ages in relation to biomass, soil properties and GHG effluxes, we used a Randomized Complete Block design in 2014. A long-term established forage study with differing stand ages was used for this experiment, and data collected for this experiment took place over one year (*Zhu et al., 1997*; *Kobayashi et al., 2018*). There were three blocks (about 2.3 ha for each block) and each block was evenly divided into four subblocks, four stand age treatments (i.e., 2, 3, 5 and 7 years old, sown in late August 2012, 2011, 2009 and 2007, respectively) were randomly assigned into each subblock. For each subblock, three sampling plots (30 m width and of 100 m length) were randomly set up for forage harvest, and soil and GHG sampling.

## Alfalfa biomass and soil property

To determine alfalfa productivity of different stage ages, one quadrat (1 m × 1 m) in each sampling plot was randomly selected and the hay yield was measured by cutting above-ground biomass during early blooming periods (10 June, 20 July and 01 October). To measure the under-ground root biomass (RBM), another quadrat of the same size in each sampling plot was randomly selected, RBM was collected by digging 30 cm depth after gas collection (see next section for details). The harvested materials were oven-dried at 60 ° C for 48 h, and then weighted.

To determine the soil characteristics in relation to field stage age, we also randomly selected two sampling sites in each sampling plot, and soil samples were collected at a 0–10 cm depth using the bucket auger (five cm diameter) after gas collection (see next section for details). Soil samples were naturally dried then extracted by passing through a 0.25-mm sieve. Soil organic carbon (SOC) was measured by Chromic acid REDOX titration (*Nelson & Sommers, 1996*). Soil total nitrogen (STN) was determined following the methods of *Bremner & Mulvaney (1982)*. Meanwhile, two cores (8.4 cm diameter × 6 cm length) were sampled by inserting soil profile of 0–10 cm depth in each quadrat and cores were dried at 105 ° C for 48 h. The soil water content (SWC) was then estimated as: (original wet weight - soil dry weight)/soil volume.

## GHG efflux

GHG effluxes from soils are more likely to occur in spring, summer and autumn than in winter (*Liu, Wang & Xu, 2010*), thus GHG samplings were only carried out during the growing seasons of April, June, July, August and October in 2014. Two sampling sites were randomly selected in each sampling plot on 13 April. Gases were sampled four times (i.e., 5:00, 10:00, 14:00 and 18:00) for three successively sunny days in each mid-month, after removing the above-ground plant and litter (*Liu et al., 2017*). The mean GHG fluxes during the three successive days were treated as the average daily fluxes for that month.

Gas was collected using a static opaque chamber (30 cm ×30 cm ×30 cm) (*Liu et al., 2017*). For each sampling event, four gas samples were taken within 30 min at a time interval of 10 min (i.e., 0, 10, 20 and 30 min). The chamber was also equipped with an electronic thermometer. The air temperature inside the chamber was recorded during gas sampling and applied to calculate gas flux (see below). Soil temperature (ST) was also measured by a mercury thermometer inserted five cm into the soil at the sampling site before and after gas sampling and the mean temperature of the two measurements was applied to detect its effect on GHG effluxes.

Gas concentration was measured within 24 h, i.e., $CH_4$ and $CO_2$ were simultaneously analyzed by a $CH_4/CO_2$ Spektrum Analyser with syringe injection (Model No. 908-0011-0001, Los Gatos Research, USA), and $N_2O$ was analyzed by a $N_2O$ Spektrum Analyser (Model No. 908-0015-0000, Los Gatos Research, USA). According to *Liu et al. (2017)*, the daily GHG fluxes were estimated as: $GHG_{daily} = (a \times flux_{7:00} + b \times flux_{12:00} + c \times flux_{16:00} + d \times flux_{18:00})$, where $a$, $b$, $c$ and $d$ are the constant gas flux duration (i.e., $a = 11$ h from 20:00 to 7:00, $b = 5$ h from 7:00 to 12:00, $c = 4$ h from 12:00 to 16:00, and $d = 4$ h from 16:00 and 20:00). The hourly GHG fluxes were thus estimated as: $GHG_{hourly} =$

GHG$_{daily}$/$e$, where $e = 24$ (number of hours per day); and the monthly GHG fluxes were then calculated as: GHG$_{monthly}$ = GHG$_{daily}$ ×$f$, where $f = 30$ or 31 (number of days per month between April and October 2014). The total gas flux during growing seasons was the sum of monthly fluxes (from April to October). Gas fluxes in May and September were not measured and thus gap-filled using linear interpolation of the arithmetical means of gas fluxes for the two close months (*Chen et al., 2013*).

The flux of GHG describes the change of gas in unit time in the sampling box. Generally, a positive value indicates gas effluxes, and a negative value suggests gas absorption. The specific formula is (*Liu et al., 2017*):

$$F = \rho \frac{V}{A} \cdot \frac{P}{P_0} \cdot \frac{T_0}{T} \cdot \frac{dCt}{dt}$$

where $F$ is the gas flux (kg/m$^2$/h), $\rho$ is the gas density (kg/m$^3$) under standard conditions ($\rho_{CO2} = 1.965$ kg/m$^3$, $\rho_{CH4} = 0.715$ kg/m$^3$ and $\rho_{N2O} = 1.965$ kg/m$^3$ respectively for CO$_2$, CH$_4$ and N$_2$O), $V$ is chamber volume (m$^3$), $A$ is the base area of the chamber (m$^2$), $P$ is the atmospheric pressure (kPa) of the sampling sites (approximately 85.48 kPa at 1,390 m above sea level), $P_0$ is atmospheric pressure under standard conditions (101.325 kPa), $T_0$ is the temperature under standard conditions (273.15 K), $T$ is the temperature (K) inside the chamber, and $dC_t/dt$ is the average rate of concentration change with time (ppm min$^{-1}$).

The total GHG efflux is estimated as the global warming potential (GWP) for a 100-year time horizon, CO$_2$-eq. One GWP of CH$_4$ accounts for 25CO$_2$-eq and one GWP of N$_2$O for 298 CO$_2$-eq (*IPCC, 2006*). Water use efficiency (WUE) was calculated according to *Sun et al. (2018)*: WUE = hay yield/(irrigation + precipitation + $\Delta$ SWC$_{October-April}$). The precipitation and irrigation from April to October 2014 was 70 and 360 mm, respectively. GHG efflux intensity measuring the ratio of GHG effluxes per unit hay yield (GEI$_{hay}$) was also estimated according to *Dyer et al. (2010)*: GEI$_{hay}$ (kg CO$_2$-eq/kg hay) = GHG efflux/hay yield.

## Statistical analysis

All other statistical analyses were conducted using SAS 9.4 (SAS Institute Inc., Cary, NC, USA). Results of a Shapiro–Wilk test (UNIVARIATE Procedure) indicated that data collected from this study were normally distributed. The difference in hay yield and WUE, and total GHG efflux (CO$_2$-eq) and GEI$_{hay}$ between different stand ages were analyzed using least significant difference test (LSD test, GLM Procedure). The correlations of soil properties (i.e., ST, SWC, SOC, STN) and RBM to GHG effluxes were determined (CORR Procedure). The variations of soil properties and RBM and hourly GHG fluxes in response to alfalfa stand age (y, year) and seasonal progress (m, month) were analyzed using a general linear model (GLM Procedure): variation = $a + b \times m + c \times m^2 + d \times y + e \times y^2 + f \times m \times y$, where $a$ is intercept, and $b$, $c$, $d$, $e$ and $f$ are estimated regression coefficients. The significant coefficients were only included in the final model. A stepwise multiple regression analysis was applied to determine the possible effects of soil properties and RBM on CH$_4$, CO$_2$ and N$_2$O fluxes (GLM Procedure) and the significant factors were only included in the final model. The proportional contributions of soil properties and

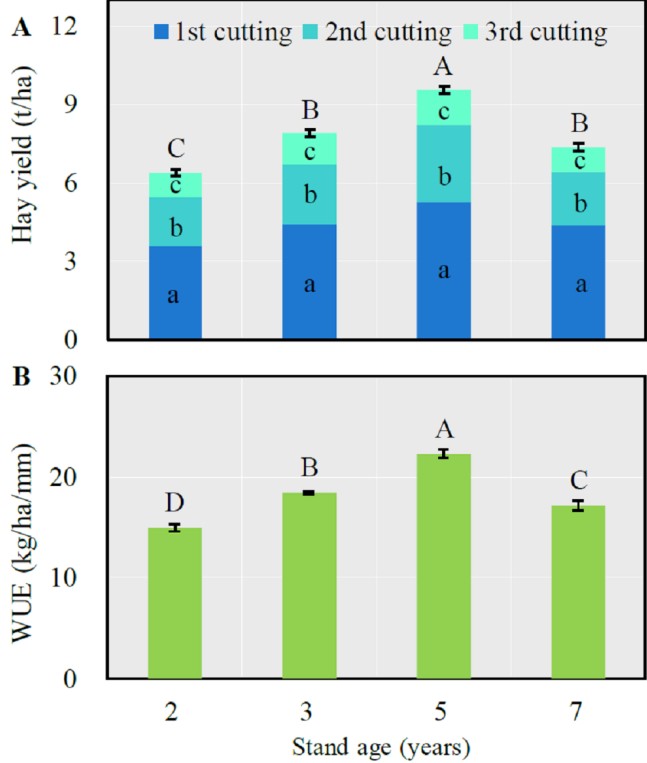

**Figure 1** **Mean (± SE) annual hay yield (A) and water use efficiency (WUE) (B) in alfalfa fields of different stand ages.** For the total hay yield (A) or WUE (B), columns with the same uppercase letters are not significantly different ($P > 0.05$). For the hay yield of each cutting, columns with the same lowercase letters are not significantly different ($P > 0.05$).

RBM to $CH_4$, $CO_2$ and $N_2O$ fluxes were then calculated as: the sum of squares for each test factor, divided by the total sum of squares then multiplied by the regression coefficient (i.e., $R^2$) of the model.

## RESULTS

### Alfalfa biomass, GHG fluxes and soil property in relation to alfalfa stand age

Both total annual hay yield and WUE significantly increased with the stand age from 2 to 5 years then significantly decreased after which time ($LSD = 0.50$ and 1.15 respectively for hay yield and WUE, $P < 0.0001$) (Fig. 1). The first cutting respectively accounted for 56.1, 55.9, 55.2 and 59.5% of total annual forage yield respectively from the 2-, 3-, 5- and 7-year-old fields, which was significantly greater than that of the second or third cutting ($P < 0.05$).

The GEI$_{hay}$ was significantly lower in 3- and 5-year-old fields than in 2- and 7-year-old ones ($LSD = 0.12$, $P < 0.0001$) (Fig. 2A), and a significantly higher annual GHG efflux was detected in 5-year-old fields ($LSD = 1.58$, $P = 0.0030$) (Fig. 2B).

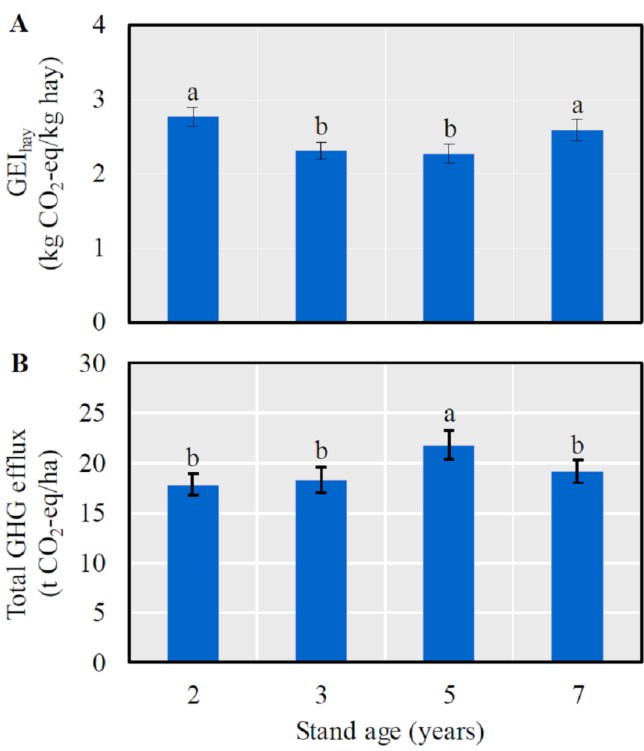

**Figure 2** **The mean (± SE) GHG efflux (CO$_2$-eq) per unit hay yield (GEI$_{hay}$) (A) and total GHG efflux (CO$_2$-eq) (B) in alfalfa fields of different stand ages.** Columns with the same lowercase letters are not significantly different ($P > 0.05$).

Alfalfa stand age had no significant effect on ST and SWC ($F_{1,56} = 1.70$ and $3.62$ respectively for ST and SWC, $P > 0.05$) (Figs. 3A–3B). However, ST significantly increased from mid-spring (April) until summer (July) ($F_{1,57} = 322.89$, $P < 0.0001$) and then significantly decreased after July ($F_{1,57} = 358.83$, $P < 0.0001$) (Fig. 3A); while a reverse seasonal pattern was detected for SWC, i.e., it significantly decreased until July ($F_{1,57} = 322.89$, $P < 0.0001$) then significantly increased ($F_{1,57} = 358.83$, $P < 0.0001$) (Fig. 3B).

Both SOC and STN increased and peaked in 5-year-old fields ($F_{1,55} = 5.28$ and $13.76$ respectively for SOC and STN, $P < 0.05$) then significantly decreased after which year ($F_{1,55} = 5.75$ and $14.08$ respectively for SOC and STN, $P < 0.05$) (Figs. 3C–3D). However, SOC significantly increased with seasonal progress and peaked in July ($F_{1,55} = 13.09$, $P = 0.0006$) after which month it significantly decreased ($F_{1,55} = 13.61$, $P = 0.0005$) (Fig. 3C); but a reverse seasonal pattern was detected for STN, i.e., it significantly decreased from April to July ($F_{1,55} = 7.78$, $P = 0.0073$) then significantly increased ($F_{1,55} = 9.35$, $P = 0.0034$) (Fig. 3D).

Both stand age and growing season initially promoted the RBM ($F_{1,54} = 268.71$ and $29.96$ respectively for stand age and month, $P < 0.0001$) (Fig. 3E). But the RBM started to decline after August ($F_{1,54} = 269.42$, $P < 0.0001$), and the decrease of RBM became slow

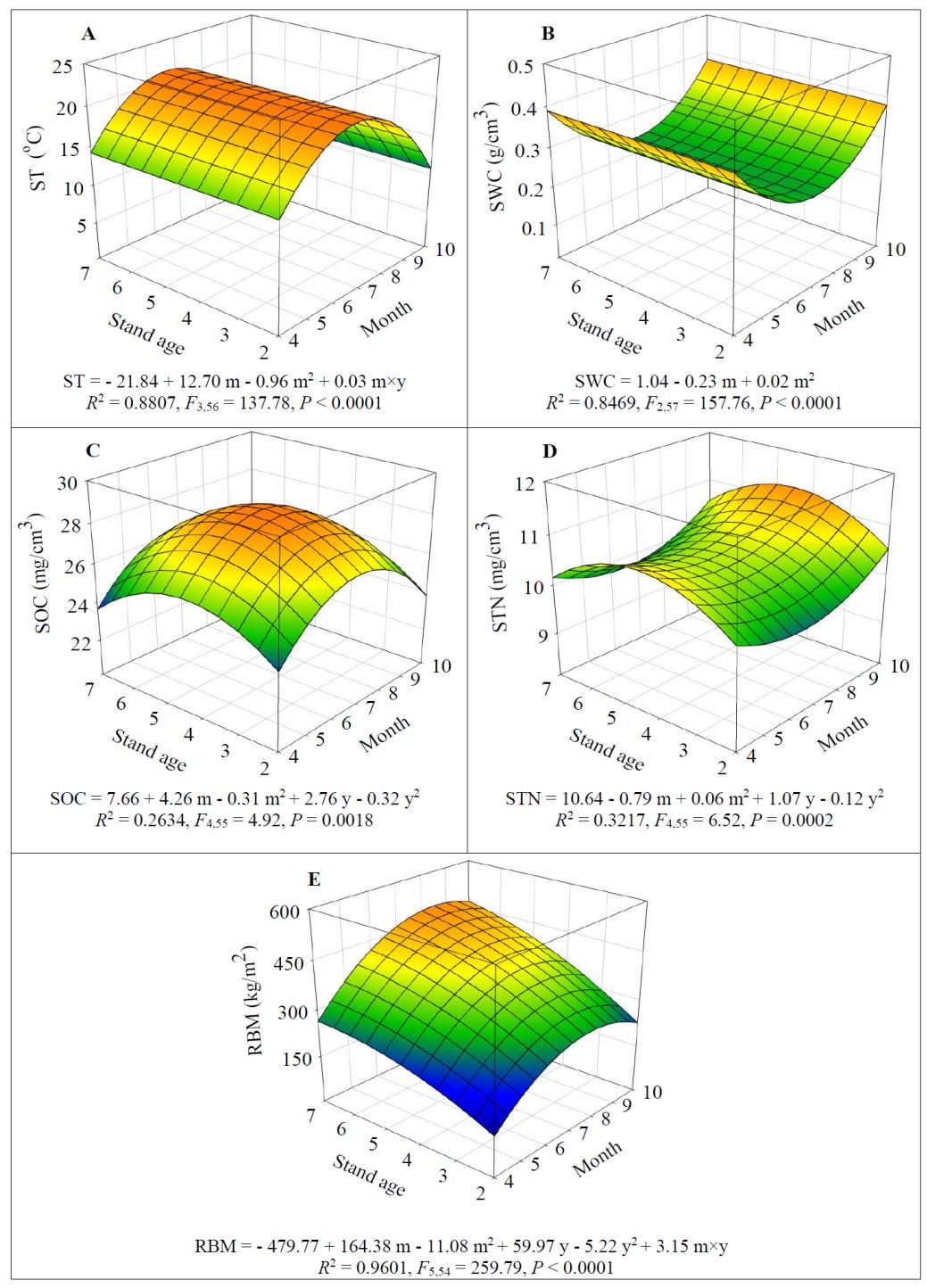

**Figure 3** Dynamics of soil temperature (ST) (A), soil water content (SWC) (B), soil organic carbon (SOC) (C), soil total nitrogen (STN) (D), and root biomass (RBM) (E) depending on alfalfa stand age (years, y) and growing season (month, m).

($F_{1,54} = 24.00$, $P < 0.0001$) due to significant positive interaction between stand age and seasonal progress ($F_{1,54} = 23.79$, $P < 0.0001$) (Fig. 3E).

The dynamics of $CH_4$, $CO_2$ and $N_2O$ fluxes also largely depended on alfalfa stand age and season (Fig. 4). $CH_4$ uptake was detected in the present study and it significantly increased when alfalfa aged up to 5 years old ($F_{1,54} = 36.24$, $P < 0.0001$) then significantly decreased ($F_{1,54} = 42.06$, $P < 0.0001$) (Fig. 4A). While $CH_4$ uptake significantly decreased from April to July ($F_{1,54} = 149.67$, $P < 0.0001$) and then significantly increased after July ($F_{1,54} = 149.05$, $P < 0.0001$) (Fig. 4A). The $CH_4$ uptake was generally higher ($-15.4$ to $-25.0$ $\mu g/m^2/h$) in 5-year-old fields for a given month.

The seasonal and annual dynamics of $CO_2$ and $N_2O$ effluxes were similar, i.e., the effluxes significantly increased when alfalfa aged to 5 years old ($F_{1,55} = 15.62$ and $15.35$ for $CO_2$ and $N_2O$ respectively, ($P < 0.001$) and the significantly decreased $F_{1,55} = 13.03$ and $13.32$ for $CO_2$ and $N_2O$ respectively, $P < 0.0001$); similarly the effluxes significantly increased since April ($F_{1,55} = 322.37$ and $195.10$ for $CO_2$ and $N_2O$ respectively, $P < 0.0001$) and then significantly decreased after July ($F_{1,55} = 363.12$ and $200.82$ for $CO_2$ and $N_2O$ respectively, $P < 0.001$) (Figs. 4B–4C). The greatest effluxes of $CO_2$ (i.e., 551.1 $mg/m^2/h$) and $N_2O$ (8.0 $\mu g/m^2/h$) were estimated in 5-year-old fields in July.

## GHG efflux in relation to soil property and root biomass

The $CH_4$ uptake significantly decreased with increasing ST and SOC but increased with increasing SWC and STN (Table 1). While $CO_2$ and $N_2O$ effluxes significantly increased with increasing ST and SOC but decreased with increasing SWC and STN (Table 1). RBM had no significant effect on $CH_4$, whereas $CO_2$ and $N_2O$ effluxes significantly increased with the increase of RBM (Table 1).

When both soil property and RBM were considered, SWC was the only factor that significantly affected $CH_4$ fluxes (Table 2). While three factors (i.e., SWC, ST and SOC) significantly affected $CO_2$ effluxes, and four factors (i.e., SWC, ST, SOC and RMB) significantly affected $N_2O$ effluxes (Table 2). SWC accounted for $\geq 65\%$ variation of $CO_2$ and $N_2O$ effluxes. ST explained about 15% variation of $CO_2$ effluxes, which was 4.3 times less than did SWC but 3.5 times more than did SOC. For $N_2O$ effluxes, ST only accounted for only $<5\%$ variation, which was 7.3 and 2.1 times less than did SWC and SOC, respectively, and RBM accounted for only $<2\%$ of variation (Table 2).

## DISCUSSION

A number of empirical studies have determined the optimal stand age of alfalfa in the semi-arid Loess Plateau and Inner Mongolia regions, while different experimental designs, field management and geographic locations could generate divergent conclusions. For example, when considering the forage yield only, the optimal stand age varies from 3 to 5 years depending on the annual precipitation (i.e., 300–500 mm) (e.g., *Du, Long & Wang, 1999b*; *Zhang et al., 2004*; *Cheng, Wan & Wang, 2005*). *Jia et al. (2009)* suggested that the optimal stand age should be 7 years if considering hay yield only but could be up to 15 years when considering WUE alone. In the present study, we show that 5 years may be the optimal alfalfa stand, for two reasons. First, both the hay yield and WUE were significantly

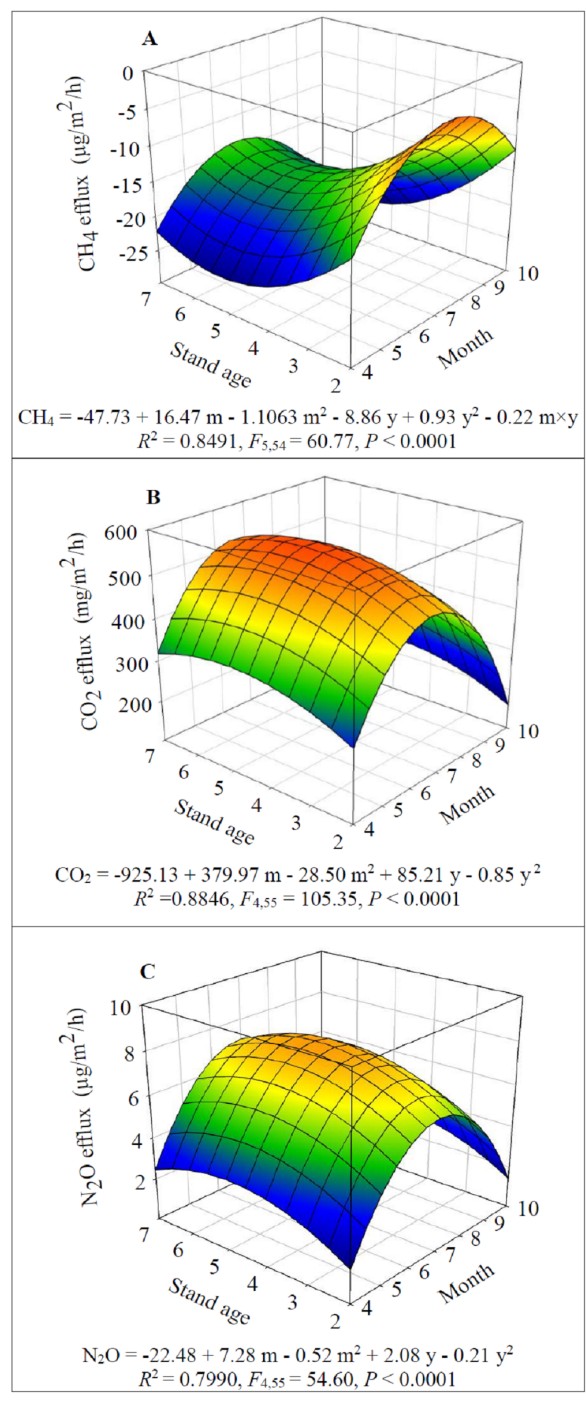

**Figure 4  Dynamics of GHG fluxes depending on alfalfa stand age (years, y) and growing season (month, m): $CH_4$ (A), $CO_2$ (B) and $N_2O$ (C).**

greater in 5-year-old alfalfa fields (Fig. 1). Second, although the significantly higher total annual GHG effluxes (Fig. 2B; also see Figs. 4B–4C), the total annual GHG efflux rate per unit hay yield (i.e., $GEI_{hay}$) was significantly lower (Fig. 2A) and the net $CH_4$ uptake

**Table 1** Soil $CH_4$ efflux ($\mu$g/m²/h), $CO_2$ efflux (mg/m² /h) and $N_2O$ efflux ($\mu$g/m²/h) correlated to soil temperature (ST, °C), soil water content (SWC, g/cm³), soil organic carbon (SOC, mg/cm³), soil total nitrogen (STN, mg/cm³) and root biomass (RBM, g/m²).

| GHG | ST | SWC | SOC | STN | RBM |
|---|---|---|---|---|---|
| $CH_4$ | 0.5027[***] | −0.6106[***] | 0.3152[*] | −0.3882[**] | −0.1533[ns] |
| $CO_2$ | 0.8756[***] | −0.8054[***] | 0.5958[***] | −0.3333[**] | 0.3852[**] |
| $N_2O$ | 0.8186[***] | −0.8200[***] | 0.6210[***] | −0.2573[*] | 0.5077[***] |

Notes.

ns, non-significant correlation

[*] <0.05.

[**] <0.01.

[***] <0.001.

**Table 2** The final optimal general linear models (GLMs) and the contribution of factors. Factors include soil temperature (ST,°C), soil water content (SWC, g/cm³), soil organic carbon (SOC, mg/cm³), soil total nitrogen (STN, mg/cm³) and root biomass (RBM, g/m²) to $CH_4$flux ($\mu$g/m²/h), $CO_2$ flux (mg/m²/h) and $N_2O$ flux ($\mu$g/m²/h).

| GHG | Factor | df | Type I SS | Contribution (%) | F | P |
|---|---|---|---|---|---|---|
| $CH_4$ | SWC | 1 | 1,104.51 | 37.28 | 34.47 | <0.0001 |
| | Error | 58 | 1,858.44 | 62.72 | | |
| $CO_2$ | SWC | 1 | 696,778.94 | 64.86 | 224.75 | <0.0001 |
| | ST | 1 | 158,944.53 | 14.80 | 51.27 | <0.0001 |
| | SOC | 1 | 44,915.24 | 4.18 | 14.49 | 0.0004 |
| | Error | 56 | 173,616.20 | 16.16 | | |
| $N_2O$ | SWC | 1 | 251.37 | 67.23 | 212.86 | <0.0001 |
| | ST | 1 | 16.38 | 4.36 | 13.87 | 0.0005 |
| | SOC | 1 | 34.43 | 9.21 | 29.15 | <0.0001 |
| | RBM | 1 | 6.75 | 1.81 | 5.72 | 0.0202 |
| | Error | 55 | 64.95 | 17.37 | | |

Notes.

Final models: $CH_4$ flux, 3.31–60.11 SWC ($R^2 = 0.3728$); $CO_2$ flux, −29.50 −473.10 SWC + 17.45 ST + 10.48 SOC ($R^2 = 0.8384$); $N_2O$ flux, −2.27 −11.98 SWC + 0.24 SOC + 0.21 ST + 0.37×10$^{-2}$ RBM ($R^2 = 0.8263$).

was higher in the 5-year-old alfalfa fields (Fig. 4A). Therefore, the net $CH_4$ sink may have largely offset the alfalfa $CO_2$ and $N_2O$ effluxes in the arid continental regions (Fig. 4). To our knowledge, this is the first study providing evidence for the assessment of optimal crop stage age integrating the total annual GHG effluxes and GEI$_{hay}$.

Alfalfa stand age affecting soil GHG effluxes is mediated by changing soil properties. Many authors have demonstrated that successive cropping alfalfa will elevate soil nutrient due to sequestering carbon (C) and nitrogen (N) into soils and eliminating tillage (*Halvorson, Wienhold & Black, 2002*; *Liang et al., 2003*). Our results show that SOC and STN continuously increased when alfalfa aged up to 5 years old (Figs. 3C–3D), which agreed with previous studies (*Xu, 2014*; *Cao et al., 2012*). Alfalfa RBM also had a similar seasonal and annual pattern as SOC (Figs. 3C and 3E). Because new root develops primarily in the spring and root biomass increases as more C is fixed by the greater leaf areas associated with plant regrowth, greater amounts of C are translocated to the root system (*Lee & Jose, 2003*; *Jiang & Claude, 2006*). However, root development and C-fixation may decline

after extended dry periods, which decreases SOC at the end of growing season (Fig. 3E). The lower STN content detected during the warm seasons may attribute to the higher uptake of soil inorganic N by the growing plants. The decreasing SOC, STN and RBM in the 7-year-old fields indicate a declining plant vitality (*Zhang et al., 2004*; *Cheng, Wan & Wang, 2005*; *Jia et al., 2009*; *Xu, 2014*) resulting in a lower forage yield (Fig. 1A).

Soil property change may influence the source and sink function of greenhouse gases (*Oertel et al., 2016*). Amount the abiotic factors, soil temperature and moisture are the two major drivers regulating GHG effluxes mainly via soil respiration and microbial activity (*Kitzler et al., 2006*; *Singh et al., 2010*; *Butterbach-Bahl et al., 2013*). As reported in a study in alfalfa fields in the dry Loess Plateau in China (*Xu, 2014*), we found that increasing soil temperature promoted $CO_2$ and $N_2O$ effluxes and suppressed $CH_4$ uptake (Table 2; Figs. 3 and 4). These results agree with the general conclusions of previous studies (e.g., *Kitzler et al., 2006*; *Singh et al., 2010*; *Butterbach-Bahl et al., 2013*; *Zhu et al., 2016*). Therefore, it may be prevalent that $CO_2$ and $N_2O$ effluxes start to increase in spring and peak in summer (Fig. 4), because the soil warming promotes soil respiration rate via microbial activity (e.g., faster growth rate and substrate use rate) (*Kitzler et al., 2006*; *Singh et al., 2010*; *Xu, 2014*).

Unlike temperature, moisture influences GHG fluxes via changing GHG diffusion rate and oxygen availability or regulating microbial communities because they require water for physiological activities (*Singh et al., 2010*). However, each soil type may have a specific soil moisture that optimizes GHG fluxes (*Schindlbacher, Zechmeister-Boltenstern & Butterbach-Bahl, 2004*). When moisture exceeds the optimum level, gas transport is restricted (*Schaufler et al., 2010*; *Kim et al., 2012*), leading to anaerobic conditions; whereas suboptimal moisture levels will limit GHG fluxes due to water stress of soil microbes (*Schindlbacher, Zechmeister-Boltenstern & Butterbach-Bahl, 2004*; *Kitzler et al., 2006*). By following this line, it may be predicted that at the arid conditions such as that of our experiment where SWC is below the optimum level, increasing SWC will promote $CH_4$ uptake and $CO_2$ and $N_2O$ emission as it elevates diffusivity of oxygen ($O_2$) in soils which is essential for soil respiration and bacterial nitrification and methanotrophy under aerobic conditions. Our results partially support the assumptions. Increased SWC induced higher $CH_4$ uptake (Figs. 3B and 4A; Tables 1 and 2), agreeing with *Dutaur & Verchot (2007)* that methanotrophy is a dominant process in upland dry soils and there is thus a net uptake of $CH_4$ by soils. However, our results show that $CO_2$ and $N_2O$ emission decreased with increasing of SWC (Figs. 3B and 4B; Tables 1 and 2). Therefore, it is supposed that different GHGs have various thresholds of SWC invoking gas emission in arid regions, which are warranted for future studies.

Beside the main abiotic drivers of soil moisture and temperature, agricultural GHG fluxes are directly mediated by biotic factors including root respiration and microbial activity (*Kitzler et al., 2006*; *Singh et al., 2010*; *Butterbach-Bahl et al., 2013*) of which are regulated by root biomass and soil nutrient such as carbon and nitrogen (e.g., *Schindlbacher, Zechmeister-Boltenstern & Butterbach-Bahl, 2004*; *Wang, Peng & Fang, 2010*; *Oertel et al., 2016*). Indeed, some researchers have reported the positive correlations between soil respiration, RBM and SOC (*Lee & Jose, 2003*; *Jiang & Claude, 2006*) and between $N_2O$ efflux and SOC (*Xu, 2014*). Our results indicate that increasing SOC and RBM significantly

elevated $CO_2$ and $N_2O$ effluxes (Table 1). According to *Xu (2014)*, the greater GHG effluxes in 5-year-old fields (Figs. 2B, 3B, and 3C) may partially attribute to the higher microbial abundance and activity and root respiration owing to the higher soil nutrient and RBM (Fig. 3).

When compared the annual dynamics of STN (Fig. 3D) with that of $N_2O$ effluxes (Fig. 4C), we may assume that increasing STN (organic and inorganic N) in soils may enhance $N_2O$ effluxes via the biological processes of nitrification or denitrification (*Xu, 2014*; *MacDonald, Farrell & Baldock, 2016*; *Oertel et al., 2016*). However, our findings do not support the above notion, rather STN had a significantly negative effect on $N_2O$ effluxes (Table 1). In agricultural systems, plants only take up inorganic N (i.e., $NO_3^--N$ and $NH_4^+-N$) (*Schmidt, Nasholm & Rentsch, 2014*), but may use organic N through the processes of mineralization (bacteria digest organic material and release $NH_4^+-N$) and nitrification (bacteria convert $NH_4^+-N$ to $NO_3^--N$) (*Schmidt, Nasholm & Rentsch, 2014*; *Fernandez & Kaiser, 2018*). The causes of negative effect of STN on N2O effluxes may be that the increasing STN (Fig. 3D) promotes the uptake of $NH_4^+-N$ and $NO_3^--N$ by plants for growth (*Ghimire, Norton & Pendall, 2013*), which reduces $NH_4^+-N$ available for microbial nitrification and results in less $N_2O$ effluxes. Considering the climate conditions in this study, the net $N_2O$ effluxes in the alfalfa fields may attribute to the processes of nitrification under aerobic conditions.

Although ST, SWC, SOC, STN and RBM alone had significant positive or negative effect on GHG effluxes (except RBM had no significant effect on $CH_4$ effluxes, Table 1), STN had little impact on GHG effluxes, and increasing RBM could significantly elevate $N_2O$ efflux, but its impact was very small, i.e., explained <2% variation (Table 2). Agreeing with that of *Schindlbacher, Zechmeister-Boltenstern & Butterbach-Bahl (2004)* and *Oertel et al. (2016)*, the flux rates of $N_2O$ and $CO_2$ largely depend on ST, SWC and SOC (Table 2). Generally, as discussed above an increase of soil temperature will lead to greater effluxes and soil respiration rates as a positive feedback response of increased microbial metabolism (*Kitzler et al., 2006*; *Singh et al., 2010*; *Butterbach-Bahl et al., 2013*; *Oertel et al., 2016*). However, *Fowler et al. (2009)* stated that the positive temperature impact could be limited by soil water stress, as water is needed as a transport medium for nutrients required by microbes. In the current study, $N_2O$ and $CO_2$ effluxes were more sensitive to SWC than to ST and SOC, and $CH_4$ efflux responded only to SWC (Table 2).

## CONCLUSIONS

Based on the forage yield, WUE, GHG efflux ($CO_2$-eq) and $GEI_{hay}$, we found that in the arid inland regions the optimal alfalfa stand age is 5 years. This knowledge is helpful in decision of alfalfa retention time based on the maximum benefit by considering the hay yield, water use efficiency, GHG emission as well as the cost of field establishment of alfalfa. Our results also indicate that in the arid regions with higher soil water stress, SWC overrides ST, SOC and RBM as a key factor regulating GHG fluxes and increasing SWC leads to an increase of net uptake of $CH_4$ by soils and a reduction of $N_2O$ and $CO_2$ effluxes from the alfalfa fields. Irrigation is required for alfalfa growing in the arid regions, while it also has

significant impacts on GHG emission (*Dutaur & Verchot, 2007*; *Schaufler et al., 2010*; *Ussiri & Lal, 2013*; *Burger et al., 2016*). Therefore, future researches on the dynamics of GHG fluxes affected by irrigation (i.e., frequency, timing, and amount of irrigation water used) are warranted to develop strategies for GHG mitigation, increasing alfalfa forage yield and prolonging alfalfa persistence in the continental arid regions.

## ACKNOWLEDGEMENTS

We thank two anonymous reviewers and the handling editor for their constructive comments on earlier versions, which have significantly improved the paper. We also are grateful to Dr Charles West (Texas Tech University) for his valuable comments and time spent editing the English of a previous version of the paper.

### Funding

This work was supported by the Strategic Priority Research Program of Chinese Academy of Science (XDA20100102), National Key Basic Research Program of China (2014CB138706), Changjiang Scholars and Innovative Research Team in University (IRT17R50) and The 111 project (B12002). The funders had no role in study design, data collection and analysis, decision to publish, or preparation of the manuscript.

### Grant Disclosures

The following grant information was disclosed by the authors:
Strategic Priority Research Program of Chinese Academy of Science: XDA20100102.
National Key Basic Research Program of China: 2014CB138706.
Changjiang Scholars and Innovative Research Team in University: IRT17R50.
The 111 project: B12002.

### Competing Interests

The authors declare there are no competing interests.

### Author Contributions

- Jiao Ning conceived and designed the experiments, performed the experiments, analyzed the data, prepared figures and/or tables, authored or reviewed drafts of the paper, and approved the final draft.
- Xiong Z. He analyzed the data, prepared figures and/or tables, authored or reviewed drafts of the paper, and approved the final draft.
- Fujiang Hou conceived and designed the experiments, analyzed the data, prepared figures and/or tables, authored or reviewed drafts of the paper, and approved the final draft.
- Shanning Lou, Xianjiang Chen, Shenghua Chang, Cheng Zhang and Wanhe Zhu performed the experiments, authored or reviewed drafts of the paper, and approved the final draft.

## Data Availability

The raw measurements are available in the Supplementary Files.

## Supplemental Information

Supplemental information for this article can be found online at http://dx.doi.org/10.7717/peerj.8738#supplemental-information.

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
