# Peer review of "Optimizing alfalfa productivity and persistence versus greenhouse gases fluxes in a continental arid region"

_PeerJ, doi:10.7717/peerj.8738_

## Round 0.1 · original submission · Major Revisions

The reviewers have some comments on the text. Overall the remarks are not very critical, however they do demand a revision. Please update the abstract and the references list. I believe you'd revise the manuscript soon, even by this year.

Reviewer 1 ·

Basic reporting

No comments.

Experimental design

No comments.

Validity of the findings

No comments.

Additional comments

This study uses a space-for-time approach to determine the optimum trade-off between alfalfa productivity and GHG-emission dynamics as related to stand age. Overall, the study is well-designed and the methods/results are clear, and the manuscript is well-written. More justification and rationale for the study is needed in the introduction, with some recommendations in the specific comments below. Additional comments/suggestions are included below.

Abstract
L32—37: Please add a sentence or two here that clearly states the implications and impact of the study results for management practices and/or ecosystem outcomes of semi-arid alfalfa stand management.
Introduction
L41 & 45: Specify that alfalfa is a forage legume, and that rhizobia are root-dwelling symbiotic bacteria.
L55: What are “certain” successive growing years? Is there a specific amount or range?
L61: The shift from alfalfa stand age in the previous paragraph to lack of alfalfa-GHG emissions research here seems sudden. Why are GHG emissions an important missing aspect of previous studies? I would recommend adding a better transition between alfalfa stands and GHG emissions, moving the third long paragraph (L68-82) about GHG emissions in general up to L59, and then expanding the paragraph currently in L59-67 to clearly explain the knowledge gap and potential ecosystem consequences of not addressing this missing research. Each of these paragraphs need clear transitions to the subsequent paragraphs/ideas.
L87: Specify which GHGs were measured and converted to CO2-equivalents.

Methods
L97-102: References for this information?
L105: The phrase “And one year study was designed due to stable climate condition and change of alfalfa stand age” followed by descriptions of different years and treatments is confusing. Does this mean that a long-term established forage study with differing stand ages was used for this experiment, and data collected for this experiment took place over one year? Please revise to clarify.
L112: “quadrat” not plural
L121: A 0.14 mm sieve seems very small. Why not 2 mm for air-dried samples? Was soil passed through this sieve size after crushing to a fine powder specifically for C and N analysis? Please clarify.
L130: Capitalize “Two”
L150: Delete “As” at beginning of sentence.
L165-166, 167-169: Could references be added for the WUE equation and GEE equation?

Results
L199: “mid-spring”?
215-218: This sentence is worded confusingly (easy to get mixed up with long sequence of “increased then decreased then decreased then increased”, etc.). Recommend splitting into at least two sentences and revising for clarity.
L223: Does “since April” mean the period leading up to April?

Discussion
L250: “although there were” instead of “although the”
L261—265: These sentences include fragments and are confusingly worded. Try altering to “…regrowth, greater amounts of C is translocated to the root system (refs). However, root development and C-fixation may decline after extended dry periods, which decreases SOC at the end of the growing season (Fig 3E).”
L289: Split into two sentences. Delete i.e. and begin with “Increased”.
L291-293: This paragraph conclusion seems sudden and could use more explanation. Why is soil structure invoked?
L324: “limited by” instead of “overlain by”?

Conclusion
L331: “overrides” instead of “overlies”?
L335—338: A much stronger study-specific conclusion and implications sentence is need after this call for more research. What are the current management recommendations and ecosystem outcomes that can be inferred from this study?

Table 2
This caption is the first time that structural equation modeling (SEM) is specifically mentioned in the manuscript. The construction and use of this model should be clearly explained in the methods and referenced in the results.

Figures 3 & 4
Recommend adding the equations/p-values for each panel to the figures themselves, not in the caption, and instead describe the parameters related within each panel.

Reviewer 2 ·

Basic reporting

See details in general comments below.

Experimental design

See details in general comments below.

Validity of the findings

See details in general comments below.

Additional comments

Dear editor and authors:

The present manuscript(ID: peerj-43325)is a report regarding optimizing alfalfa productivity and persistence vs GHG fluxes in the arid area of China. The reviewer argue this study with good design and labored works to give some novel information for accessing GHG fluxes of alfalfa production. The overall MS was prepared well except for those figures with poor quality, hopefully, it could be improved if this MS could go further. In addition, some suggestions and questions could be considered for revising as follow:
-With respect to abstract, I would suggest to rewrite it, please highlight the objective of study and add information about treatments.
-Line 69 and 74- Carbon dioxide and methane could be changed to CO2 and CH4.
-Line 131 to 133-is that ok using mean gases records from three successively sunny days to represent the average daily fluxes for that month? According to your results regarding SWC, I assume that gases fluxes would be changed largely when the field was irrigated, if it is necessary to record more measurements during that time. Definitely, you could not do it again, thus please add a reference here.
-Line 132-“after removing the above-ground materials”, remove the plants or something else, please make it clear.
-Line 134-Please add information about the size of chamber.
-Line 168 to 169-Please add the information on GEEwater.
-Line 170 to 185- suggest to remove all fig and table information from this part.
-Line 193 to 194- no statistical analysis results were shown in the fig, add or revise the description.
-Line 215- not clear “it increased with increasing when alfalfa”, please revise.
-Line 283-change leads to leading.
--Line 312- fig.
-Line 312-313-“may be that the increasing STN (Fig. 3D) promotes the uptake of NH4+-N plants, producing higher forage yield (Fig. 1A)”, how does increasing STN promote the uptake of NH4-N by plants? dose alfalfa prefer to use NH4+-N rather than NO3-? Please explain the causes of negative effect of STN on N2O effluxes with more accurate words.
-As for reference, suggest using more documents published recently.


Based on above comments, the final recommendation regarding this manuscript is major revision.

---

## Round 0.2 · accepted · Accept

The paper has been improved at last revision. There are no more critical remarks. Please check again the English grammar (at the proofreading stage).

Reviewer 1 ·

Basic reporting

no comment

Experimental design

no comment

Validity of the findings

no comment